# S-Propargyl-Cysteine Ameliorates Peripheral Nerve Injury through Microvascular Reconstruction

**DOI:** 10.3390/antiox12020294

**Published:** 2023-01-28

**Authors:** Haiyan Xi, Chenye Wang, Qixiu Li, Qing Ye, Yizhun Zhu, Yicheng Mao

**Affiliations:** 1Shanghai Key Laboratory of Bioactive Small Molecules, School of Pharmacy, Fudan University, Shanghai 201203, China; 2School of Pharmacy, Macau University of Science and Technology, Macau, China

**Keywords:** S-propargyl-cysteine, hydrogen sulfide, microvascular reconstruction, peripheral nerve injury repair, sirtuin1, Notch1 intracellular domain

## Abstract

Microvascular reconstruction is essential for peripheral nerve repair. S-Propargyl-cysteine (SPRC), the endogenous hydrogen sulfide (H_2_S) donor, has been reported to promote angiogenesis. The aim of this study is to utilize the pro-angiogenic ability of SPRC to support peripheral nerve repair and to explore the potential mechanisms. The effects and mechanisms of SPRC on angiogenesis and peripheral nerve repair were examined under hypoxic condition by establishing a sciatic nerve crushed injury model in mice and rats, and a hypoxia model in human umbilical vascular endothelial cells (HUVECs) in vitro. We found that SPRC accelerated the function recovery of the injured sciatic nerve and alleviated atrophy of the gastrocnemius muscle in mice. It facilitated the viability of Schwann cells (SCs), the outgrowth and myelination of regenerated axons, and angiogenesis in rats. It enhanced the viability, proliferation, adhesion, migration, and tube formation of HUVECs under hypoxic condition. SPRC activated sirtuin1 (SIRT1) expression by promoting the production of endogenous H_2_S, and SIRT1 negatively regulated Notch signaling in endothelial cells (ECs), thereby promoting angiogenesis. Collectively, our study has provided important evidence that SPRC has an effective role in peripheral nerve repair through microvascular reconstruction, which could be a potentially effective medical therapy for peripheral nerve injury.

## 1. Introduction

Peripheral nerve injury caused by trauma or compression leads to sensory and motor dysfunction. Its symptoms include numbness, tingling, throbbing, burning, and sharp pain. These injuries may lead to lifelong loss of sensory and motor functions, incurring substantial financial costs and diminished quality of life, which is one of the most prominent health-related problems and an important medical issue but lacks satisfying treatment [1,2,3]. Unlike the central nerves system, the peripheral nerves system exhibits some self-healing potential after mild and moderate trauma, but this is mainly based on its regenerative microenvironment, including microvascular reconstruction. Blood vessels provide injured nerves with oxygen and nutrients, exchange heat, help remove wastes produced in the regeneration process, and offer a physical link between nerve stumps as tracks to direct Schwann cells (SCs) migration across the wound [4]. The loss of vascular reconstruction has a negative impact on axons and myelin regeneration, and also up-regulates the inflammation level in regenerative microenvironment [5]. However, most studies on revascularization are based on tissue engineering therapies, such as supplementing with proangiogenic factors or incorporating endothelial cells (ECs) and stem cells, and it is currently lacking the effective drugs needed to achieve revascularization [6,7,8,9].

S-Propargyl-cysteine (SPRC) is an endogenous hydrogen sulfide (H_2_S) donor. Our previous studies have demonstrated that SPRC promotes angiogenesis in hindlimb ischemia model of mice and myocardial ischemia model of rats by up-regulating cystathionine-γ-lyase (CSE)—the enzyme producing endogenous H_2_S [10]. The pro-angiogenic ability of SPRC hints that it could potentially be a therapeutic drug candidate to promote microvascular reconstruction, thus could further support the repair and regeneration of peripheral nerve injury.

Sirtuin1 (SIRT1) is a conserved protein nicotinamide adenine dinucleotide (NAD^+^)-dependent deacetylase, which regulates various physiological functions, such as apoptosis, cell senescence, gene transcription, energy balance, oxidative stress, and metabolic regulation, by catalyzing deacetylation of key proteins [11,12,13,14]. It has been reported that SIRT1 controls endothelial angiogenic functions during vascular growth by regulating signaling pathways via its deacetylase activity. Loss of SIRT1 function blocks embryonic angiogenesis and branch morphogenesis of ECs [15]. In ECs, SIRT1 is a key mediator of pro-angiogenic signals secreted from myocytes and loss of SIRT1 leads to decreased vascular density in skeletal muscle [16]. SIRT1 is regulated by a variety of endogenous molecules, diet and environmental stress [11,13], and its expression is down-regulated under hypoxia [17]. Evidence indicates that H_2_S could up-regulate the expression of SIRT1 [18,19,20].

Considering the important role of SIRT1 in angiogenesis and the regulatory effect of H_2_S on SIRT1, we attempted to use the endogenous H_2_S donor SPRC as a regulator of SIRT1 to promote microvascular reconstruction under hypoxic condition after peripheral nerve injury. NaHS, an exogenous donor of H_2_S, served as a control for the study. In addition, DL-propargylglycine (PAG), a specific inhibitor of CSE, and EX527, a specific inhibitor of SIRT1, were added in the in vitro study to prove the rationality of the study, respectively.

The Notch pathway has been reported to be involved in multiple aspects of vascular development, such as proliferation, migration, and angiogenesis. However, whether Notch regulates vascular development positively or negatively is still debated [21], though the involvement of Notch1 intracellular domain (NICD) in angiogenesis as the deacetylation substrate of SIRT1 has been previously reported [22]. In this study, we intended to further verify the interaction between SIRT1 and NICD during angiogenesis under hypoxic condition. 

In short, the aim of this study was to utilize the pro-angiogenic ability of SPRC to support peripheral nerve repair and to explore the potential mechanisms. We found that SPRC accelerated the function recovery of the injured sciatic nerve and alleviated atrophy of the gastrocnemius muscle in mice. It facilitated the viability of SCs, the outgrowth and myelination of regenerated axons, and angiogenesis in rats. It enhanced the viability, proliferation, adhesion, migration, and tube formation of human umbilical vascular endothelial cells (HUVECs) under hypoxic condition. SPRC activated SIRT1 expression by promoting the production of endogenous H_2_S, and SIRT1 negatively regulated Notch signaling in ECs under hypoxic condition, thereby promoting angiogenesis.

## 2. Materials and Methods

### 2.1. Reagents

SPRC (purity > 99%) was synthesized, as reported in previous study [23]. NaHS, PAG, and EX527 were purchased from Sigma-Aldrich (Darmstadt, Germany). The drug actions are shown in Table 1. The cell culture and assay reagents were purchased from Meilunbio (Dalian, China) and Servicebio (Wuhan, China). Primary and secondary antibodies for western blotting (WB), immunohistochemistry, and immunofluorescence were purchased from Abcam (Cambridge, UK) and Proteintech (Wuhan, China). NaHS and SPRC were freshly prepared before each experiment.

### 2.2. Experimental Animals

Adult male C57BL/6J mice (8 weeks old, weight 20–22 g) and Sprague-Dawley rats (8 weeks old, weight 200–220 g) were purchased from Lingchang Biotech (Shanghai, China) and housed in the Experimental Animal Center of Fudan University at constant temperature (25 ± 2 °C) and humidity (50–70%) in a 12-h light/dark cycle, and allowed food and water freely. All animal procedures were approved by the Ethical Committee on Animal Experiments (School of Pharmacy, Fudan University).

### 2.3. Sciatic Nerve Crush Injury Procedure and Drug Treatments

The mice and rats were randomly divided into four groups (5 mice and 5 rats per group): sham group (sham surgery + 0.9% saline), model group (sciatic nerve crush injury + 0.9% saline), NaHS group (sciatic nerve crush injury + 2.8 mg/kg/day NaHS treatment), SPRC group (sciatic nerve crush injury + 50 mg/kg/day SPRC treatment). Unilateral crush injury was performed on the sciatic nerve of animals. All animals were anesthetized with 1–2% isoflurane. After anesthesia, the skin in the left thigh area was sterilized. The left sciatic nerve was exposed after surgical incision of overlying skin and muscles and crushed by a toothless hemostatic forceps for 30 s (the degree of crush was three teeth bitten in the handles) at the site 1 cm proximal to the main bifurcation for the tibial and peroneal nerves. Then the clamping site was marked with a 10-0 microscopic suture under aseptic condition. Finally, the fascia, subcutaneous tissue, and skin were sutured layer-by-layer. The animals in the sham group were subjected to the surgery described above but not to nerve injury. After surgery, animals were kept on a heating plate at 37 °C until they had recovered completely from anesthesia.

The animals in the NaHS and SPRC groups were treated with NaHS (2.8 mg/kg/day, diluted in sterile 0.9% saline) or SPRC (50 mg/kg/day, diluted in sterile 0.9% saline) by intraperitoneal injection for 2 weeks, and the animals in the sham and model groups were given an equal volume of 0.9% saline by intraperitoneal injection. At 2 weeks after surgery, animals were euthanized via CO_2_ asphyxiation and relevant assessments were performed.

### 2.4. Sciatic Function Index (SFI) Measurement

Before euthanasia, gaits of mice were analyzed in each group for computing the SFIs. The CatWalk XT 10.6 system (Noldus Inc., Wageningen, the Netherlands) was used to assess gait recovery and motor function. This test involved monitoring each animal when it crossed a walkway with a glass floor illuminated along the long edge. Data acquisition was carried out using a high-speed camera, and pawprints were automatically classified by the software. The performance of each mouse was recorded three times to obtain approximately 15-step cycles per mouse for analysis. Next, two variables, with respect to the pawprints of the experimental lateral foot (E) and normal lateral foot (N), were measured: (1) print length (PL): the distance between the third toe and the heel; (2) toe spread (TS): the distance between the first and fifth toes. The above two variables were entered into the following formula to calculate the SFI. SFI = 118.9 (ETS − NTS/NTS) − 51.2 (EPL − NPL/NPL) − 7.5. SFI = 0 was regarded as normal, and SFI = −100 indicated complete nerve damage [24].

### 2.5. Gastrocnemius Muscle Assessment

Gastrocnemius muscle wet weight measurement, hematoxylin-eosin (HE) staining, and fiber diameter quantification were conducted to observe the pathological changes of atrophy. Bilateral gastrocnemius muscles of mice were harvested and weighed. The wet weight ratio (%) was calculated as (the wet weight of the injured side/the wet weight of the normal side) × 100%.

For HE staining, the gastrocnemius muscles were cut into 1 cm^3^ blocks transversely in the center of the muscle, fixed with 4% paraformaldehyde (PFA) solution, dehydrated with ethanol solutions, and then embedded with paraffin. Cross-sectional 4 µm thick slices of the muscle were prepared and stained with HE. Images were obtained with VS200 ASW system (Olympus, Tokyo, Japan) and the maximum transverse diameter of each fiber was quantified by ImageJ software 1.8.0 (NIH, Bethesda, MD, USA). The quantitative analysis was performed with the mean value of each muscle-fiber maximum transverse diameter.

### 2.6. Sciatic Nerve Histological Assessment

Immunofluorescence was applied to evaluate the levels of recovery and regeneration of sciatic nerve. The axonal regrowth and SCs activity in rats were evaluated by the expressions of a neurofilament 200 (NF200) and S100 calcium binding beta protein (S100β), respectively. The 5 mm segments of injured sciatic nerve were removed, fixed with 4% PFA solution, permeabilized with 5% Triton X-100 solution, and incubated with 5% bovine serum albumin (BSA) solution. Subsequently, the samples were incubated overnight with anti-NF200 (1:200) and anti-S100β (1:200) antibodies. The next day, the samples were treated with CoraLite488-conjugated goat anti-mouse antibodies (1:200) and CoraLite594-conjugated goat anti-rabbit antibodies (1:200). Then the samples were coverslipped using antifade mounting medium with 4′,6-diamidino-2-phenylindole (DAPI, a DNA-specific fluorescent probe). Images were obtained by a confocal microscope (Olympus, Tokyo, Japan), and the NF200-positive number and the S100β-positive area were quantified by ImageJ software 1.8.0.

### 2.7. Sciatic Nerve Morphological Assessment

Ultrastructural analyses of axon density, axon diameter, and myelination thickness were conducted in rats by semi-thin slices and ultra-thin slices to observe the regeneration of axon and myelin sheaths. The 1 mm^3^ segments of injured sciatic nerve from rats were fixed in 2.5% glutaraldehyde, then fixed in 1% osmium tetroxide, dehydrated with ethanol solutions, and embedded in acetone and Epon 812 resin (1:1) for 30 min. After additional infiltration in Epon 812 for 2 h, the samples were cut into 1 µm semi-thin sections and mounted on slides, which were dyed with 1% toluidine-blue (TB)-staining. Images were obtained through VS200 ASW system. In addition, embedded samples were sliced into 50 nm ultra-thin sections and placed on 200 mesh copper grids and stained with uranium acetate and lead nitrate for 30 min. Sections were observed and photographed by transmission electron microscope (TEM).

### 2.8. Vascularity of Sciatic Nerve Assessment

Immunohistochemistry was applied to counting capillaries in injured sciatic nerves of rats. The injured sciatic nerve was fixed with 4% PFA, paraffin-embedded, and cut into cross-sections with a 4 µm thickness. The sections were incubated with anti-platelet ECs adhesion molecule (PECAM-1/CD31) antibody and then stained with an appropriate secondary antibody conjugated with horseradish peroxidase. Images were obtained with VS200 ASW system and the CD31-positive area was counted by ImageJ software 1.8.0.

### 2.9. Cell Culture and Treatments

HUVEC line was purchased from the Chinese Academy of Science Cell Bank (Shanghai, China). HUVECs were cultured with Dulbecco’s modified Eagle’s medium (DMEM) containing 10% fetal bovine serum (FBS) and 1% penicillin-streptomycin, and incubated in a humidified incubator with 5% CO_2_ at 37 °C. Cultured HUVECs were divided into six groups: normoxia group (normoxia + vehicle), hypoxia group (hypoxia + vehicle), NaHS group (hypoxia + 50 µM NaHS treatment), SPRC group (hypoxia + 50 µM SPRC treatment), PAG group (hypoxia + 50 µM SPRC + 1 mM PAG treatment), and EX527 group (hypoxia + 50 µM SPRC + 10 µM EX527 treatment). Stock solution of reagents were prepared in dimethyl sulfoxide (DMSO) and subsequently diluted in PBS. Cells in the hypoxia and treatment groups were exposed in hypoxic condition with 1% O_2_, 5% CO_2_ and 94% N_2_ for 6–24 h at 37 °C. Cells in the normoxia group were cultured in a 5% CO_2_ and 95% air at 37 °C.

### 2.10. Cell Viability, Proliferation, Adhesion, Migration and Tube Formation Assay

The cell viability of HUVECs was assessed by a Cell Counting Kit-8 (CCK-8) assay (Beyotime, Shanghai, China). HUVECs were inoculated in 96-well plates at a density of 5 × 10^3^ cells/well and subjected to different treatment for 24 h, 10 µL/well CCK-8 solution was added to the culture media and incubated for 2 h. Finally, the absorbance values at 450 nm were detected by a microplate reader (Thermo Fisher Scientific, Waltham, MA, USA) and analyzed by ImageJ software 1.8.0.

The cell proliferation of HUVECs was assessed by BeyoClick™ EdU cell proliferation kit with alexa fluor 488 (Beyotime, Shanghai, China). HUVECs were inoculated into 48-well plates at a density of 5 × 10^3^ cells/well and treated as indicated above. The cells were then incubated with EdU (5-ethynyl-2′-deoxyuridine, a thymidine analog that can be incorporated into replicating DNA for detection of cell proliferation) for 2 h according to the manufacturer’s instructions. Subsequently, the cells were fixed in 4% PFA and stained with Hoechst 3334. Images were obtained by a fluorescence microscope (Olympus, Tokyo, Japan) and the EdU-positive cells were calculated from three randomly selected fields by ImageJ software. The EdU-positive cells ratio (%) was calculated as the (EdU-positive cells/Hoechst 3334-positive cells) × 100%.

The adhesion of HUVECs was assessed through the adhesion assay. The wells of the 96-well plate were coated with 20 µL/well of Matrigel (10 mg/mL, Corning Inc., Corning, NY, USA), and 2 × 10^4^/well pretreated HUVECs were seeded on the gel and incubated for 10 min. The nonadherent cells were washed away with PBS. The adherent cells were fixed with 4% PFA and stained with 0.5% crystal violet. Three randomly selected fields were captured by an inverted microscope (Olympus, Tokyo, Japan) and the adherent cells were calculated by ImageJ software 1.8.0.

The migration of HUVECs was assessed through the wound healing assay (horizontal migration) and transwell chamber assay (vertical migration). For the wound healing assay, cells were inoculated in 6-well plates and scraped with a 200 µL sterile micropipette tip, and the floating cells were washed with PBS. The HUVECs were treated as indicated above. Wound recovering was observed and photographed by an inverted microscope at 0 h and 24 h. The percentage of wound closure was calculated as the relative ratio of the migration distance by ImageJ software 1.8.0. For transwell chamber assay, a total of 1 × 10^5^ pretreated cells in 100 µL of serum-free growth medium were inoculated into the upper chamber with 8 µm pores and 500 µL of complete media with 10% FBS and different treatment was added to the lower compartment of 24-well. After incubation for 24 h, non-migrated cells on the upper surface of the upper chamber were removed with cotton buds and migrated cells in the bottom of the well were fixed with 4% PFA and stained with 0.5% crystal violet. Images were obtained by an inverted microscope and cells in three random fields were counted by ImageJ software 1.8.0.

The in vitro angiogenic ability of HUVECs was tested by tube formation assay using Matrigel. The wells of the 96-well plate were coated with 50 µL/well of Matrigel. HUVECs were starved in DMEM without serum for 24 h prior to the assay, and then 2 × 10^4^/well pretreated HUVECs were inoculated on the gel. The cells were incubated with different treatment for 6 h. Representative photos of the tube structure were taken by an inverted microscope and the branch points of blood vessels were measured by ImageJ software 1.8.0.

### 2.11. H_2_S Level and NAD^+^/NADH (the Reduced Form of NAD^+^) Ratio Measurement

The injured sciatic nerve of rats, or 5 × 10^6^ cells/sample HUVECs (different treatment for 6 h), were collected, and the intracellular H_2_S level and NAD^+^/NADH ratio were determined by a Micro H_2_S content assay kit (Solarbio, Beijing, China) and a NAD^+^/NADH assay kit with WST-8 (Beyotime, Shanghai, China) according to the manufacturer’s instructions. For H_2_S level, the principle of assay is that H_2_S reacts with N, N-dimethyl-p-phenylenediamine, zinc acetate and ferric ammonium sulfate to form methylene blue, which has a maximum absorption peak at 665 nm. The content of H_2_S can be calculated by measuring its absorbance value. The samples were lysed with lysis buffer and centrifuged, then the supernatant was collected and added to detection reagents. Subsequently, the absorbance values were measured at 665 nm. H_2_S concentration was calculated by the standard curve from manufacturer. For NAD^+^/NADH ratio, the samples were lysed with lysis buffer and centrifuged, and the supernatant was collected. To measure total NAD^+^ and NADH, 20 µL of the supernatant was added to a 96-well plate. To measure NADH, the supernatant was incubated at 60 °C for 30 min and 20 µL was added to a 96-well plate. Subsequently, 90 µL of alcohol dehydrogenase was added and incubated at 37 °C for 10 min. Finally, 10 µL of chromogenic solution was added to the plate and the mixture was incubated at 37 °C for 30 min. Standard curve was generated and measured at the same time as the samples. The absorbance values were measured at 450 nm. The amount of NAD^+^ was derived by subtracting NADH from total NAD^+^ and NADH. The resultant value was normalized to the ratio of NAD^+^/NADH.

### 2.12. Immunofluorescence and Co-Immunoprecipitation (Co-IP) of SIRT1 and NICD

After HUVECs were exposed to normoxia or hypoxia for 6 h, immunofluorescence staining for SIRT1 and NICD was performed as described in Section 2.6. The anti-SIRT1 (1:200) and anti-NICD (1:200) antibodies were used for SIRT1 and NICD colocalization, respectively. After exposure to hypoxia for 6 h, HUVECs were lysed in IP lysis buffer and centrifuged, then the supernatant was collected and precleared with 40 µL the protein A/G magnetic beads for 1 h. Immunoprecipitation was performed by adding the primary anti-SIRT1 antibody to the precleared supernatant and then rocking overnight. This was paired with the species-matched mouse IgG as a negative control. The antigen-antibody complex was incubated with 50 µL protein A/G agarose beads for 2 h. After washing 4 times with the IP buffer, the immunoprecipitation was resuspended in a 1 × sodium dodecyl sulfate (SDS) sample buffer and boiled at 95 °C for 5 min for later WB analysis.

### 2.13. Western Blotting Analysis

The samples were lysed in RIPA buffer and the total proteins were quantified using the bicinchoninic acid (BCA) method. Electrophoresis was performed using 10% SDS polyacrylamide gel. After blotting the proteins with polyvinylidene fluoride (PVDF) membrane, blocking was performed for 2 h using 5% BSA solution. Then membranes were incubated overnight at 4 °C with the following primary antibodies: β-tubulin mouse mAb (1:5000), CSE mouse mAb (1:5000), SIRT1 mouse mAb (1:5000), NICD rabbit mAb (1:1000), Delta-like ligand 4 (DLL4) rabbit mAb (1:1000), Jagged1 rabbit mAb (1:1000), and hairy and enhancer of split 1 (Hes1) rabbit mAb (1:1000). After a reaction with the primary antibodies, the samples were incubated with secondary antibodies (anti-mouse IgG2 and anti-rabbit IgG2 at 1:5000 dilution) solution for 2 h. Then the immune complexes were detected using enhanced chemiluminescence (ECL) solution, and the grey values of each band were determined using ImageJ software 1.8.0.

### 2.14. Statistical Analyses

Data were expressed as the mean values ± standard error of the mean (SEM). For the cell experiments, the data for each assay were obtained from the average of three independent experiments. For the animal experiments, the data for each assay were obtained from the average of five animals in each group. Multigroup comparisons were performed with one-way ANOVA followed by Tukey’s multiple comparison test using GraphPad Prism version 9.0. Statistical significance was set at *p* < 0.05.

## 3. Results

### 3.1. SPRC Accelerated the Motor Function Recovery of Injured Sciatic Nerve and Alleviated the Atrophy Degree of Gastrocnemius Muscle in Mice

All animals remained in good health during the course of experimentation and no animals suffered an infection or failed surgical wound healing. Gait analysis was performed at 2 weeks after surgery in mice to evaluate the recovery of motor function. The representative pawprints and SFI of each group are shown in Figure 1A. During free walking, the five toes of each healthy mouse had to maintain a certain distance and a stable grip on the ground. After the sciatic nerve was damaged, the five toes were unable to separate, and the toes stayed tight. After 2-week treatment, the five toes of the NaHS and SPRC groups separated, indicating early recovery of the treated mice. The SFI value of the NaHS group was higher than that of the model group (*p* < 0.01) and the SFI value of the SPRC group was even more significant (*p* < 0.001), but both were lower than that of the sham group, consistent with the results from pawprints.

The sciatic nerve controls the gastrocnemius muscle. Therefore, the functional regeneration of the sciatic nerve can be reflected by the degree of atrophy of the gastrocnemius muscle. After the sciatic nerve is damaged, the gastrocnemius muscle becomes atrophy and its weight decreases, and muscle fibers shrink. Gross observation of gastrocnemius muscles and statistical analysis of wet weight ratio of gastrocnemius muscles showed that the atrophy degree of the gastrocnemius muscle in the NaHS group was lighter than that in the model group (*p* < 0.01), and SPRC was even more conducive (*p* < 0.001) (Figure 1B). The images of HE staining of gastrocnemius muscle and statistical analysis of quantification of gastrocnemius fibers diameter showed that the gastrocnemius fibers in the NaHS and SPRC groups were larger in diameter than that of model group (*p* < 0.01 and *p* < 0.001), indicating that NaHS and SPRC were beneficial to relieve the atrophy of gastrocnemius muscle during nerve regeneration process in mice, and SPRC was more conducive (Figure 1C). 

### 3.2. SPRC Facilitated the Viability of SCs, the Outgrowth and Myelination of Regenerated Axons, and Angiogenesis in Rats

Immunofluorescence of NF200 and S100β, ultrastructural analyses of axon and myelination thickness, and immunohistochemistry of CD31 were conducted in rats to observe the viability of SCs, the regeneration of axon and myelin sheaths, and vascularity of sciatic nerve in rats at 2 weeks after surgery (Figure 2). Immunofluorescence staining for NF200, a marker for myelinated A-fibers [25], showed that the numbers of regenerated axons in the NaHS and SPRC groups were higher than those in the model group (*p* < 0.001). S100β, a marker of SCs, is related to the formation of myelin sheath, and the results revealed that there were more activated SCs in the NaHS and SPRC groups than those in model group (*p* < 0.001) (Figure 2A). TB staining of semi-thin sections revealed that the numbers of myelinated nerve fibers were higher in the NaHS and SPRC groups when compared with the model group. TEM images of ultra-thin sections and the local enlargement of the myelin sheath further indicated that the numbers of myelinated nerve fibers were higher in the NaHS and SPRC groups, and the myelin sheath thickness of the regenerated nerve fibers in the SPRC group was greater than that in the model group, but seemingly no difference was observed between the NaHS and model groups (Figure 2B). Orange arrows indicated the regenerated axons. 

Angiogenesis improves nutrient and oxygen supply into the damage site. CD31, a marker of EC adhesion molecule, was used to verify the influence of NaHS and SPRC on angiogenesis in the process of nerve regeneration. The microvessel density was assessed by immunostaining for CD31. Poor intraneural angiogenesis was in the model group and more capillaries were in the NaHS and SPRC groups. Quantitative analysis of the vascular area confirmed that the NaHS and SPRC groups endowed the formation of significantly more vessels than model group (*p* < 0.001), attesting that NaHS and SPRC stimulated intraneural neovascularization (Figure 2C).

### 3.3. SPRC Facilitated the Viability, Proliferation, Adhesion, Migration and Tube Formation of HUVECs under Hypoxic Condition In Vitro

To investigate the effects of SPRC on the angiogenic potential of HUVEC in vitro, HUVECs were subjected to different treatment according to the experimental design, and viability, proliferation, adhesion, migration, and tube formation assay were assessed (Figure 3). The results of CCK-8 assay showed that SPRC promoted HUVECs viability under hypoxia in a dose-dependent manner after treatment for 24 h. The viability of cells treated with 50 µM or 100 µM SPRC significantly increased (*p*  < 0.001). However, the effect of 100 µM SPRC was not superior to that of 50 µM SPRC, which seemed to be the optimal dose. In addition, the viability of cells treated with 50 µM NaHS or 50 µM SPRC for 24 h obviously increased (*p* < 0.001), but co-treatment with the CSE inhibitor PAG or the SIRT1 inhibitor EX527 counteracted the effect of SPRC (Figure 3A). The results of EdU staining assay, adhesion assay, wound healing assay, transwell chamber assay, and tube formation assay showed that SPRC and NaHS effectively promoted HUVECs proliferation, adhesion, migration and tube formation. However, the positive effects of SPRC on HUVECs were attenuated by PAG and EX527 (Figure 3B–F).

### 3.4. SPRC Facilitated Angiogenesis under Hypoxic Condition In Vitro through the H_2_S/SIRT1/NICD Signaling

After demonstrating the angiogenic capability of NaHS and SPRC in vitro, we attempted to further explore the potential mechanism behind this important phenomenon. NaHS and SPRC are two different types of H_2_S donor, and NaHS can directly produce exogenous H_2_S, while SPRC generates endogenous H_2_S by increasing the expression of CSE. After 6 h of NaHS or SPRC treatment under hypoxia, the level of H_2_S in HUVECs increased significantly (*p* < 0.001). The addition of PAG inhibited the increase in H_2_S, but EX527 did not significantly inhibit H_2_S level (*p* < 0.05) (Figure 4A). SIRT1 is a NAD^+^ dependent deacetylase, and the NAD^+^/NADH ratio is significantly correlated with SIRT1 activity [26]. After 6 h of NaHS or SPRC treatment, the ratio of NAD^+^/NADH in HUVEC increased (*p* < 0.01), and the use of inhibitor EX527 made the ratio increase more significantly (*p* < 0.001), while PAG played an inhibitory role (Figure 4B).

To verify the correlation of location between SIRT1 and NICD, after being treated with normoxia or hypoxia for 6 h, HUVECs were double immunostained with SIRT1 (green) and NICD (red) antibodies. Under normoxic condition, SIRT1 mainly distributed in the cytoplasm and NICD mainly distributed in the nucleus. However, under hypoxic condition, SIRT1 had nuclear translocation and co-located with NICD mainly in the nucleus (Figure 4C). The co-IP results further confirmed the interaction between SIRT1 and NICD. The co-IP results demonstrated that NICD protein was detected in the anti-SIRT1 immunoprecipitated complexes rather than the IgG control (Figure 4D). These results have indicated that SIRT1 co-localizes and interacts with NICD under hypoxic condition.

The proteins from the HUVECs were purified for WB at 6 h after appropriate treatments to evaluate the expressions of CSE, SIRT1, NICD, DLL4 (a ligand for the Notch receptors, is a negative-feedback regulator that may block ECs proliferation and restrain vessel branching during sprouting angiogenesis [27,28]), Jagged1 (another ligand for the Notch receptors, is a potent proangiogenic regulator [29]), and Hes1 (a transcription factor for downstream target gene of the Notch pathway) with β-tubulin as a loading control (Figure 4E). Treatment of HUVECs with NaHS or SPRC under hypoxic conditions increased the expressions of CSE and SIRT1 and decreased the expression of NICD (*p* < 0.001). Co-treatment with the CSE inhibitor PAG or the SIRT1 inhibitor EX527 eliminated the SIRT1 activation and NICD inhibition caused by SPRC. It was found that NaHS or SPRC treated HUVECs increased the expression of Jagged1 and decreased the expression of DLL4 under hypoxic condition (*p* < 0.01). Co-treatment with PAG or EX527 eliminated the up-regulation of Jagged1 and down-regulation of DLL4 by SPRC. It was also found that NaHS or SPRC could significantly up-regulate the expression of Hes1 (*p* < 0.001), and both inhibitors also inhibited this effect. These results indicated that SPRC activated SIRT1 expression by producing H_2_S, while SIRT1 negatively regulated Notch signaling in HUVECs under hypoxic condition.

### 3.5. SPRC May Promote Angiogenesis in Injured Sciatic Nerve of Rats In Vivo through the H_2_S/SIRT1/NICD Signaling

The potential mechanisms of NaHS and SPRC angiogenesis were demonstrated in vitro, followed by validation in injured sciatic nerves of rats. After 2 weeks of NaHS or SPRC treatment, the level of H_2_S in injured sciatic nerves increased significantly (*p* < 0.001) (Figure 5A). However, compared with the model group, the ratio of NAD^+^/NADH in injured sciatic nerve did not show significant changes (Figure 5B). The proteins from the injured sciatic nerves of the rats were purified for WB at 2 weeks after surgery to evaluate the expressions of CSE, SIRT1, NICD with β-tubulin as a loading control (Figure 5C). Treatment of rats with NaHS or SPRC after sciatic nerve crush injury increased the expressions of CSE and SIRT1 and decreased the expression of NICD (*p* < 0.001).

## 4. Discussion

Peripheral nerves are highly vascularized tissues with intrinsic and extrinsic vessels. Peripheral nerve injury is often accompanied by the vascular injury. Ischemia and hypoxia are considered to be important factors leading to failure of regeneration of peripheral nerve, such as ischemia/hypoxia-induced inflammation promoting nerve scar formation and causes neuropathic pain [30,31]. Therefore, rapid and adequate vascular network reconstruction is a critical measure for repair and regeneration of peripheral nerve.

Many studies have reported that endogenous H_2_S plays a critical role in angiogenesis by many means, such as up-regulating the expression of vascular endothelial growth factor (VEGF) and contributing to angiogenic signals in response to VEGF [32,33]. By controlling the production of endogenous, H_2_S has become an important study direction for the treatment of angiogenesis related diseases. In this study, we found that SPRC significantly stimulated intraneural neovascularization in vivo, and facilitated angiogenesis under hypoxic condition in vitro. In previous reports, the dose and way of administration of SPRC varied in different disease models [23,34,35,36]. In this study, we found that intragastric (oral) administration caused gastrointestinal symptoms, such as diarrhea in individual animals, but no significant adverse effects occurred with intraperitoneal injection. Through pre-experiments, we found that the dose of 50 mg/kg/day SPRC can achieve good therapeutic effects on sciatic nerve injuries in rats and mice (data not shown). Moreover, the efficacy and safety of SPRC were superior to NaHS, the exogenous H_2_S donor which is very unstable [37] and irritating, and may cause pulmonary inflammation [38] or bronchitis [39] when administered intraperitoneally to animals. 

In the mechanistic of action exploration, we found that SPRC increases the expression of CSE to produce endogenous H_2_S, which may activate SIRT1 by increasing the NAD^+^/NADH ratio, and SIRT1 attenuates the inhibitory effect of Notch signaling pathway on angiogenesis by deacetylating NICD under hypoxic condition (Figure 6). SIRT1 negatively regulated Notch signaling in ECs, which was consistent with previous reports [22]. It has been reported that H_2_S acts as an inducer of nicotinamide phosphoribosyltransferase (NAMPT) which serves as the rate-limiting enzyme to convert nicotinamide (NAM) to nicotinamide mononucleotide (NMN). The NMN is then converted to NAD^+^ by NAMPT. Inhibition of CSE significantly decreased the expression of NAMPT and intracellular the level of NAD^+^ [40,41]. Whether H_2_S increases the NAD^+^/NADH ratio by the same mechanism under hypoxia needs to be further validated in the future. However, we did not observe the significant change of NAD^+^/NADH ratio in the injured sciatic nerve after 2 weeks of NaHS or SPRC treatment, which may be related to the active biological role of NAD^+^ in the organism. The generation and consumption of NAD^+^ and NADH are controlled by multiple complex pathways [42,43]. In addition, our detection time was at 2 weeks after NaHS or SPRC treatment, when obvious angiogenesis was observed (Figure 2C), which was different from the hypoxic state in the early stage of sciatic nerve injury. It might be able to explain the unchanged NAD^+^/NADH ratio in vivo.

The Notch ligands DLL4 and Jagged1 have opposing effects on angiogenesis, and Jagged1 is proangiogenic and functions by down-regulating DLL4-Notch signaling [29]. We found that SPRC up-regulated the expression of Jagged1 and decreased, but did not completely inhibit, the expression of DLL4 in HUVECs under hypoxic condition. Moderate inhibition of DLL4 can promote the increase of vascular density, but DLL4 deficiency may cause vascular defects, such as disrupted vascular hierarchy, and reduced vessel caliber [44]. Hes1 has been reported to be highly involved in angiogenesis [45,46]. In this study, we found that Hes1 was up-regulated after treatment of SPRC. 

In the sciatic nerve, besides vascular ECs, there are neurons and a large number of SCs and other components, and verification of the mechanism of SPRC promoting angiogenesis on injured sciatic nerves will be interfered by other components. Although we found that the expressions of CSE and SIRT1 were up-regulated and the expression of NICD was down-regulated in the sciatic nerve after SPRC treatment, these changes were the collective manifestation of the total proteins of the injured sciatic nerve, and it was impossible to distinguish whether the changes were from ECs, SCs or other components, after all, SCs make up the largest proportion in the sciatic nerve. It has been reported that CSE expression in SCs is up-regulated after sciatic nerve injury and is associated with myelin fragmentation, axonal degradation, SCs dedifferentiation, and proliferation [47], which suggests that SPRC may have a direct effect on the repair of injured sciatic nerve. This needs to be confirmed in future studies.

The inflammatory response to peripheral nerve injury might lead to neuropathic pain and modulating the inflammatory response to reduce persistent pain is one therapeutic challenge for peripheral nerve injury [48,49]. Previous studies have shown that SPRC exerts anti-inflammatory effects in various inflammatory conditions [50,51,52]. In this study, we speculated that SPRC may have anti-inflammatory effects in addition to stimulating intraneural neovascularization during peripheral nerve repair after injury. This also needs to be confirmed in future studies.

## 5. Conclusions

In conclusion, this study has shown that SPRC has a therapeutic effect on nerve regeneration and functional recovery. It promotes angiogenesis in injured peripheral nerves. SPRC activated SIRT1 expression by producing endogenous H_2_S, and SIRT1 negatively regulated Notch signaling under hypoxic condition in ECs, thereby promoting angiogenesis in injured peripheral nerves. Our study confirms the feasibility of revascularization of injured peripheral nerves through drug treatment, and our findings suggest that SPRC represents a novel drug therapeutic approach for peripheral nerve injury.

## Figures and Tables

**Figure 1 antioxidants-12-00294-f001:**
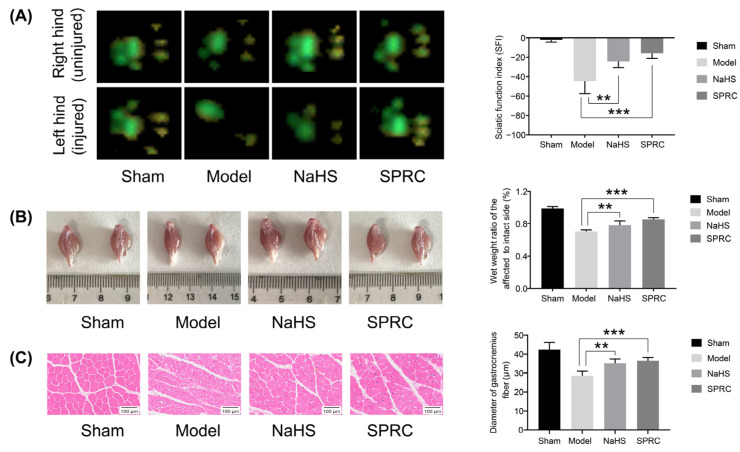
Gait analysis and gastrocnemius muscle assessments in mice at 2 weeks after surgery for each group. (**A**) Representative pawprint and statistical analysis of the sciatic function index (SFI). (**B**) Gross observation of injured (left) and uninjured (right) gastrocnemius muscles, and statistical analysis of wet weight ratio (the weight of the injured side divided by the uninjured side) of gastrocnemius muscles. (**C**) Cross sections images of gastrocnemius muscle by hematoxylin-eosin (HE) staining, and statistical analysis of quantification of gastrocnemius fibers diameter. Data were presented as mean ± SEM (*n* = 5 mice). ** *p* < 0.01 vs. model group, *** *p* < 0.001 vs. model group.

**Figure 2 antioxidants-12-00294-f002:**
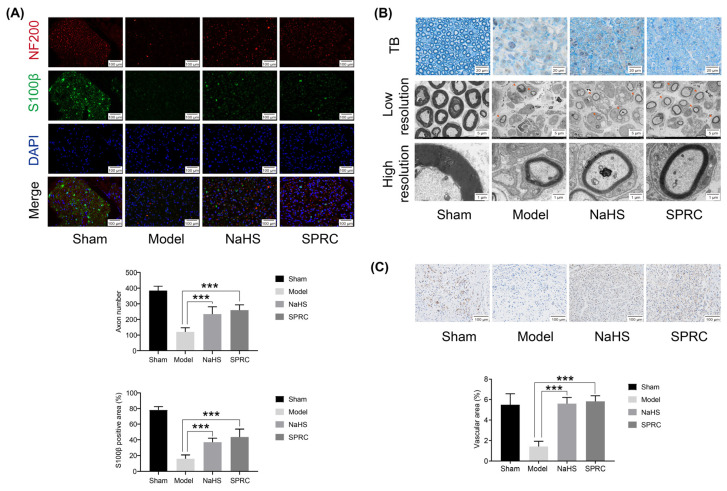
Axon regeneration, Schwann cells (SCs) viability and angiogenesis assessments in rats at 2 weeks after surgery for each group. All images came from the middle cross-sections of the injured nerves. (**A**) Representative images of neurofilament 200 (NF200) and S100 calcium binding beta protein (S100β) immunofluorescence staining, and statistical analyses of axon number and S100β positive area. NF200 (red) indicated the axons, S100β (green) indicated the SCs, and DAPI (blue) showed the nucleus. (**B**) Toluidine-blue (TB) staining of semi-thin sections and transmission electron microscope (TEM) images of ultra-thin sections. Orange arrows indicated the regenerated axons. (**C**) Representative images of platelet endothelial cell adhesion molecule (PECAM-1/CD31) immunohistochemistry staining and statistical analysis of the vascular area. Data were presented as mean ± SEM (*n* = 5 rats). *** *p* < 0.001 vs. model group.

**Figure 3 antioxidants-12-00294-f003:**
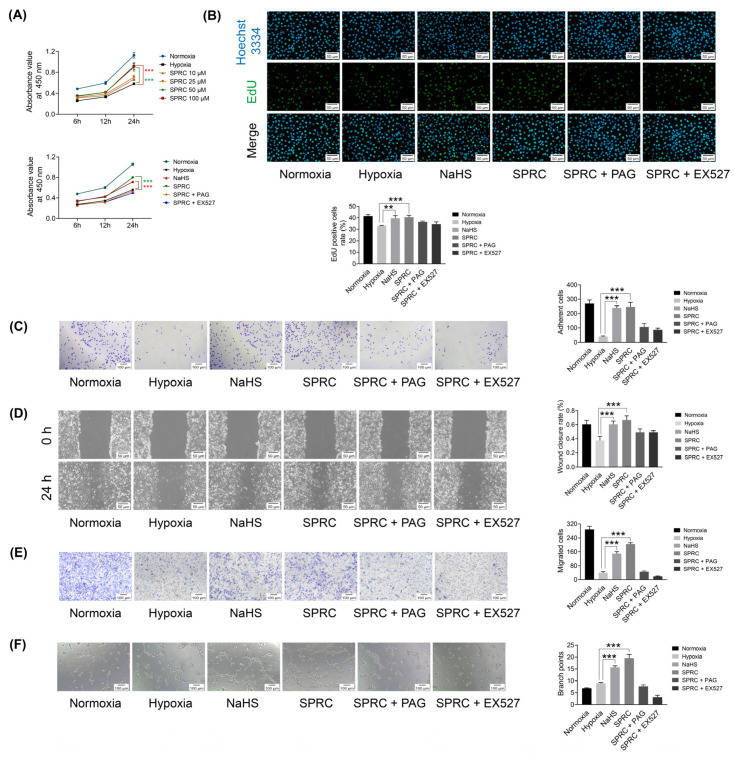
Human umbilical vascular endothelial cells (HUVECs) viability, proliferation, adhesion, migration and tube formation assessments in vitro under hypoxic condition for each group. (**A**) Statistical analysis of HUVECs viability at 24 h after different treatments. The absorbance values were measured at 450 nm. (**B**) Representative images of EdU staining and statistical analysis of EdU-positive cells rate (EdU-positive cells divided by Hoechst 3334-positive cells) at 24 h after different treatments. EdU (green) indicated the proliferating cells and Hoechst 3334 (blue) showed the nucleus. (**C**) Representative images of HUVECs adhesion and statistical analysis of adherent cells at 24 h after different treatments. (**D**) Representative images of wound healing at 0 h and 24 h after different treatments, and statistical analysis of wound closure rate (the migration distance at 24 h divided by the cell-free clear distance at 0 h). (**E**) Representative images of transwell migration and statistical analysis of migrated cells at 24 h after different treatments. (**F**) Representative images of the tube-forming and statistical analysis of branch points at 6 h after different treatments. Data were presented as mean ± SEM (*n* = 3 independent experiments). ** *p* < 0.01 vs. hypoxia group, *** *p* < 0.001 vs. hypoxia group.

**Figure 4 antioxidants-12-00294-f004:**
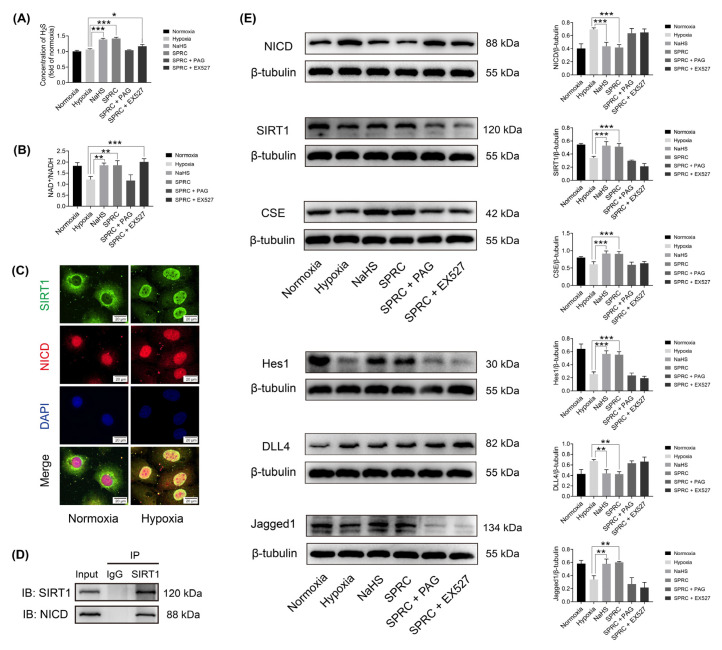
Hydrogen sulfide (H_2_S) level, nicotinamide adenine dinucleotide (NAD^+^)/NADH ratio, sirtuin1 (SIRT1) and Notch1 intracellular domain (NICD) cell localization, interaction, and cystathionine-γ-lyase (CSE), SIRT1, NICD, Delta-like ligand 4 (DLL4), Jagged1 and hairy and enhancer of split 1 (Hes1) protein expression assessments in HUVECs at 6 h after different treatments. (**A**) Statistical analysis of intracellular H_2_S level. (**B**) Statistical analysis of intracellular NAD^+^/NADH ratio. (**C**) Co-localization of SIRT1 (green) and NICD (red) under normoxic or hypoxic condition. DAPI (blue) showed the nucleus. (**D**) Co-immunoprecipitation of NICD and SIRT1 under hypoxic condition. IP, immunoprecipitation; IB, immunoblotting. (**E**) Western blotting (WB) analysis of CSE, SIRT1, NICD, Jagged1, DLL4 and Hes1 protein expressions. Relative grey values analyses were performed. β-tubulin was used as a loading control. Data were presented as mean ± SME (*n* = 3 independent experiments). * *p* < 0.05 vs. hypoxia group, ** *p* < 0.01 vs. hypoxia group, *** *p* < 0.001 vs. hypoxia group.

**Figure 5 antioxidants-12-00294-f005:**
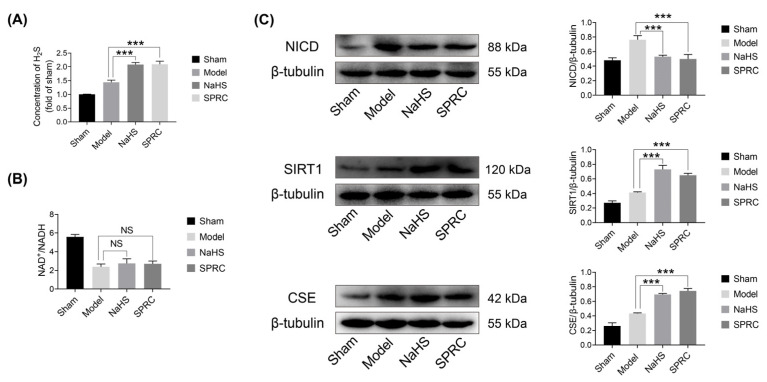
H_2_S level, NAD^+^/NADH ratio, CSE, SIRT1 and NICD protein expressions assessments in injured nerves of rats at 2 weeks after surgery for each group. (**A**) Statistical analysis of H_2_S level. (**B**) Statistical analysis of NAD^+^/NADH ratio. (**C**) WB analyses of CSE, NICD and SIRT1 protein expressions. Relative grey values analyses were performed. β-tubulin was used as a loading control. Data were presented as mean ± SEM (*n* = 5 rats). *** *p* < 0.001 vs. model group, NS, no significant.

**Figure 6 antioxidants-12-00294-f006:**
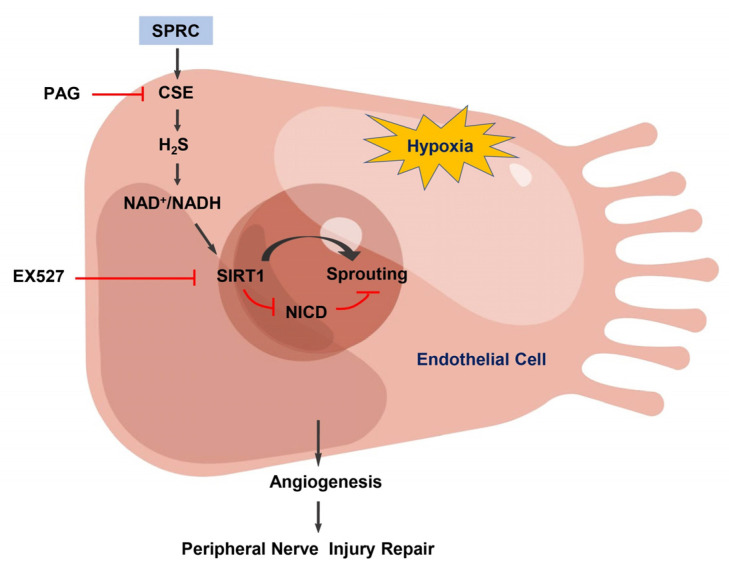
Schematic illustration of H_2_S/SIRT1/NICD-mediated mechanism in SPRC-induced angiogenesis under hypoxic condition. SPRC increases the expression of CSE to produce endogenous H_2_S, and H_2_S may activate SIRT1 by increasing the NAD^+^/NADH ratio. Under hypoxia, SIRT1 deacetylated NICD and relieved the inhibitory effect of Notch signaling pathway on angiogenesis.

**Table 1 antioxidants-12-00294-t001:** List of the drug actions used in this study.

Drug	Action
S-Propargyl-cysteine (SPRC)	an endogenous hydrogen sulfide (H_2_S) donor
NaHS	an exogenous H_2_S donor
DL-Propargylglycine (PAG)	a specific inhibitor of cystathionine-γ-lyase (CSE)
EX527	a specific inhibitor of sirtuin1 (SIRT1)

## Data Availability

The data presented in this study are contained within the article and available on request.

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
