# Peer review of "S-Propargyl-Cysteine Ameliorates Peripheral Nerve Injury through Microvascular Reconstruction"

_antioxidants, 2023, doi:10.3390/antiox12020294_

Round 1

Reviewer 1 Report

The authors investigated about the effects of S-propargyl-cysteine on angiogenesis and peripheral nerve repair. The rational behind the study was clear and straight forward. The manuscript is almost well written. Overall the topic could be interesting but some details could be improved.

 I recommend that the paper be accepted with minor revision:

a)  The authors should mentioned in the abstract more details about model used.

b)  In the introduction section, little previous evidence is provided about the importance of peripheral nerve injury in daily life. Incorporating comparisons with other studies would increase the strength of the paper. Please refer to doi: 10.3390/ijms21155330; 10.3389/fneur.2019.0063.

c)  The authors should clarify why they use this dose of SPRC  (50mg/kg) and why this method of administration (i.p.)? Any references?

d)   The authors should better emphasize the conclusions

e)   There are some minor grammar issues that should be fixed in order to aid the accessibility of the results to the reader.

Reviewer 2 Report

In the present study it was demonstrated that S-propargylcysteine (SPRC) improved injury-induced repair of sciatic nerve in the rat and mice models as well as increased regenerating nerve vascularization. In addition, SPRC stimulated angiogenic potential of HUVECs cultured in hypoxic conditions by up-regulating SIRT1 and inhibiting NOTCH signaling.

1)     Lines 46-47: it is stated that SPRC exerts its protective effects by increasing the expression of CSE. Although increase in CSE expression was observed after CSE administration, this compound is primarily the CSE substrate (doi: 10.1007/978-3-319-18144-8_16).

2)     Line 246: “the reduce form” should be corrected to: “the reduced form”.

3)     Section 2.11: the principle of H2S assay should be described in 2-3 sentence.

4)     Statistical analysis: it is stated that each experiment was repeated at least three times. What does it mean specifically? There were 5 animals in each group.

5)     The conclusion that SPRC and NaHS improve nerve repair by inducing angiogenesis is the over-interpretation of the findings; there is no clear evidence that nerve repair was mediated by angiogenesis. H2S donors could have angiogenesis-independent effects on peripheral nerves as well.

6)     Did PAG and EX527 inhibit protective effects of NaHS similarly to the effects of SPRC? Was SIRT1 involved in the effect of SPRC?

7)     What is the mechanism through which SPRC and NaHS increased NAD+/NADH ratio?

8)     SPRC-induced increase in CSE could be explained by the adaptation to increased substrate availability. However, what is the mechanism through which NaHS improves the expression of CSE in HUVECs?

9)     Line 474: the sentence finished with “its vascular” is not stylistically correct.

10) Line 486: what is the evidence of improved safety of SPRC over NaHS in this study?

Reviewer 3 Report

Hai-Yan Xi and coworkers examine the effects of S-propargyl-cysteine (SPRC; H2S donor) on angiogenesis and peripheral nerve repair in mouse and rat sciatic nerve crushed injury models (Figs. 1 and 2) and molecular mechanisms for these effects in hypoxia models in human umbilical vascular endothelial cells (HUVECs; Figs. 3 and 4) and also in injured rat sciatic nerves (Fig. 5).  The experiments were performed by using the biochemical, histological, immunohistochemical, Western blotting techniques and so on.  As a result, SPRC was found to accelerate injured sciatic nerve function recovery and alleviate gastrocnemius muscle atrophy in mice, and to facilitate Schwann cell viability, regenerated axon outgrowth and myelination, and angiogenesis in rats.  Moreover, SPRC was found to enhance HUVEC viability, proliferation, adhesion, migration and tube formation under hypoxia, and to activate SIRT1 expression by increasing H2S production in injured rat sciatic nerves.  SIRT1 regulated Notch signaling in endothelial cells in a negative manner, leading to angiogenesis promotion.  It was concluded that SPRC plays a role in peripheral nerve repair by reconstructing microvaculature and may be useful in a medical therapy for peripheral nerve injury.  This manuscript is well written and seems to be interesting.  There are several points that should be addressed and may serve to amend this manuscript, as follows:

Major points:

1.     This manuscript is a little difficult to read, because many abbreviations are used.  In order to aid in understanding this manuscript, the authors should provide an abbreviation list of the words used and a list of the drug actions mentioned here.

2.     It is unclear why mice are used in Fig. 1 while rats in Figs. 2 and 5.  Please make this point clear.

3.     The scaling bar in the figures (Figs. 1C, 2A, 2B, 2C, 3B-F, 4C) is thin and hard to see.  This point should be amended.

4.     The size of characters on the vertical and horizontal axes of the bar graph in the figures (Figs. 3-5) is very small.  This point should be amended.

Specific points:

1.     Line 88: it will not be necessary to repeatedly define “SPRC”.

2.     Line 164: “DAPI” should de defined.

3.     Lines 192 and 193: please delete one of two “streptomycin”.

4.     Lines 196 and 197: please state why the concentrations of PAG and EX527 were used.

5.     Line 205: “kit” will not be necessary.

6.     Line 208: does “United States” mean “USA”? (please see line 151)

7.     Line 213: please explain shortly “Edu”.

8.     Line 218: please insert “Adhesion assay” into a sentence.

9.     Lines 231 and 232: please alter “μl” to “μL”.  “μL” should be used throughout the text.

10.  Line 268: “respectively” should be put here.

11.  Line 276: “minutes” should be “min” (please see line 257).

12.  Line 281: please expand “PVDF”.

13.  Line 339: is “NF200” a marker of myelinated A fibers?

14.  Line 376: “24h” should be “24 h”.

15.  Line 380: there is not explanation about “PAG” and “EX527” (please see lines 427 and 428).

16.  Line 406: is it OK that SPRC increases the expression of CSE?  Please see lines 488 and 489.

17.  The fourth paragraph on page 10: “DLL4”, “Jagged1” and “Hes1” should be explained where they appear first.

18.  Lines 453 and 454: “rats” should be put in this subtitle.

19.  Lines 512 and 513: the explanation of Hes1 has been given twice (see lines 436 and 437).  Please amend this point.

20.  Lines 517-519: the sentence given here is not complete.  Please amend this point.

Round 2

Reviewer 2 Report

The manuscript has been revised according to the reviewers' comments.

I have no further concerns.

Author Response

We truly appreciate your comments. We have revised our manuscript according to other reviewer’s suggestions. Please refer to the revised manuscript.

Reviewer 3 Report

This revised manuscript has been largely amended according to my comments, and there is no concern in the present manuscript except for the following minor comments:

1.     Lines 2, 10, 61 and Table 1: not “S-propa..” but “S-Propa..”?

2.     Lines 27 and 239: not “eagle” but “Eagle”?

3.     Lines 32 and 95: not “DL-Propa..” but “DL-propa..”?

4.     Line 206 not “triton” but “Triton”?

5.     Line 217: “analysis” should be “analyses”,

6.     Line 249: please put a space between “CO2” and “and”.

7.     Line 307: please put a space between “665” and “nm”.

8.     Line 351: “was” should be “were”.

9.     Lines 406, 413 and 415: “number” should be “numbers”.

10.  Line 417: “were” should be “was”.

11.  Line 529: “(e)” should be “(E)”.

12.  Line 542: “evaluated” should be “evaluate”.
